# GRFormer: Grouped Residual Self-Attention for Lightweight Single Image Super-Resolution

Anonymous

## ABSTRACT

Previous works have shown that reducing parameter overhead and computations for transformer-based single image super-resolution (SISR) models (e.g., SwinIR) usually leads to a reduction of performance. In this paper, we present GRFormer, an efficient and lightweight method, which not only reduces the parameter overhead and computations, but also greatly improves performance. The core of GRFormer is Grouped Residual Self-Attention (GRSA), which is specifically oriented towards two fundamental components. Firstly, it introduces a novel grouped residual layer (GRL) to replace the QKV linear layer in self-attention, aimed at efficiently reducing parameter overhead, computations, and performance loss at the same time. Secondly, it integrates a compact Exponential-Space Relative Position Bias (ES-RPB) as a substitute for the original relative position bias to improve the ability to represent position information while further minimizing the parameter count. Extensive experimental results demonstrate that GRFormer outperforms state-of-the-art transformer-based methods for x2, x3 and x4 SISR tasks, notably outperforming SOTA by a maximum PSNR of 0.23dB when trained on the DIV2K dataset, while reducing the number of parameter and MACs by about **60%** and **49%** in only self-attention module respectively. We hope that our simple and effective method that can easily applied to SR models based on window-division self-attention can serve as a useful tool for further research in image super-resolution. The code is available at https://github.com/sisrformer/GRFormer.

## 1 INTRODUCTION

Single Image Super-Resolution (SISR) aims to enhance image resolution by reconstructing a high-resolution image from a low-resolution counterpart. With the development of CNN-based and transformer-based SR models, one achievement after another has been achieved for single image super-resolution tasks. For whether CNN-based models or transformer-based ones, it is an easy way to improve performance by increasing the number of the network layers and feature dimension, accompanied by the increase of parameters and calculations. A straight question should be: Is it possible to improve the performance while reducing number of the parameters and calculations for transformer-based SR models? Motivated by this question, we conduct in-depth research into self-attention and present three sub-questions concerning self-attention:

**Unpublished working draft. Not for distribution.**

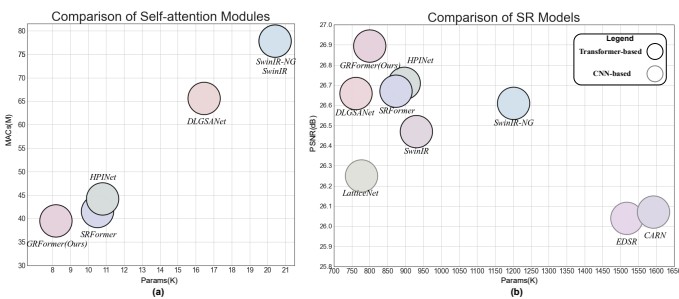

**Figure 1: (a) shows the comparisons of self-attention of recent transformer-based SR models in terms of multiply-accumulate operations (MACs) and parameters. (b) shows SISR comparisons of recent SR models (×4) in terms of PSNR on Urban100, network parameters. Our model (GRFormer) outperforms the SOTA model (x4) by 0.19dB in PSNR score while having comparably low network parameters and MACs.**

- **RQ1.** Is there any redundancy within self-attention?
- **RQ2.** Can the expressive power of self-attention be further improved?
- **RQ3.** Is there a better alternative to relative position bias for representing the position information?

**The research into RQ1.** Self-attention [20] has achieved great performance in the fields of text, image, and video since it was published in 2017. However, despite the effectiveness of self-attention mechanisms, they are often criticized for their extensive parameter count and computational demands. Existing work [3] has observed redundancy in self-attention layers, but its solutions focus mainly on how to reduce the size of the attention window [21]. Through empirical analysis, we explore the interaction among the varying parameter counts, MACs, and the performance of the self-attention module, as depicted in Fig. 1. This analysis reveals significant potential for optimization in both parameters and computational efficiency within the SwinIR's self-attention mechanism. Inspired by these findings, we propose a novel grouping scheme of Q, K, V linear layer, aimed at reducing both the parameter overhead and computational complexity.

**The research into RQ2.** In the field of single image super-resolution, previous transformer-based approaches [12, 22] have primarily utilized residual learning [8] at the outer layers of self-attention mechanisms. This methodology has been effective in mitigating network degradation and improving the expressive capabilities of deep networks. Given the success of residual learning in enhancing network performance, an intriguing question arises: could the integration of residual connections within the Query, Key, and Value (QKV) linear layers of self-attention mechanisms further augment their expressive power? Motivated by this consideration, our work

explores the incorporation of residual connections directly into the QKV linear layers, aiming to enhance their representational efficacy in deep neural architectures.

**The research into RQ3.** When self-attention [20] is proposed, it uses position embedding to provide position information of words in text. After self-attention is applied to computer vision, relative position bias (RPB) is found to be more suitable for representing relative position information, which is important for models in computer vision. However, RPB has four fatal flaws: First, if we assume the shape of the window in self-attention is H × W, the number of parameters occupied by RPB is (2H-1)×(2W-1), which is a huge parameter overhead. Second, the original RPB in SwinIR[12] sets a position parameter for each position within the window, which is not only the redundancy of the parameters, but also it is easy to be interfered by noises during training. Specifically, many subtle fluctuations can be found in the figures of Fig. 2 (a). Third, during training, each parameter of RPB is trained independently, ignoring the relative relationship between the weights of different positions. Fourth, RPB fails to clearly reflect intuition: For reconstructing an image, near pixels tend to be more important than far pixels. Motivated by this, we designed an Exponential-Space Relative Position Bias (ES-RPB) to replace RPB.

First and foremost, to solve the **RQ1**, we propose a grouping scheme for the QKV linear layer. Furthermore, to solve the **RQ2** and compensate for performance loss from the grouping scheme, we add residuals for the QKV linear layer. Lastly, to solve **RQ3**, we propose Exponential-Space Relative Position Bias (ES-RPB). The three methods above constitute the two fundamental components of Grouped Residual Self-Attention (GRSA). Given the proposed GRSA, we design a lightweight network for SR, termed GRFormer. We evaluate our GRFormer on five widely-used datasets. Benefiting from the proposed GRSA, our GRFormer achieves significant performance improvements on almost five benchmark datasets. Notably, trained on DIV2K dataset[19] for x2 SR task, our GRFormer achieves a 33.17 PSNR score on the challenging Urban100 dataset[9]. The result is much higher than recent SwinIR-light[12](32.76) and the SOTA lightweight SR model (32.94). This improvement is consistently observed across x3 and x4 tasks. Comprehensive experiments show that GRFormer not only outperforms previous lightweight SISR models [12, 14, 22], but also reduces the parameter count by about **20%** in total model architecture, compared with SwinIR [12] (1000k parameters) with the same hyperparameter. To sum up,our contributions can be summarized as follows:

- We propose a novel Grouped Residual Self-Attention (GRSA) for lightweight image super-resolution, which can not only reduces the parameter count but also enhances the performance in an easy-to-understand way for SR tasks. In addition, our proposed GRSA module can seamlessly replace the self-attention module and its variants in other transformer-based SR models, simultaneously reducing the number of parameter by about 60% and the number of MACs by about 49% in only self-attention module.
- Based on GRSA, we construct a novel transformer-based SR network, termed GRFormer. Our GRFormer achieves state-of-the-art performance in lightweight image SR, and outperforms previous lightweight SISR networks by a large margin in most of the five benchmark datasets.

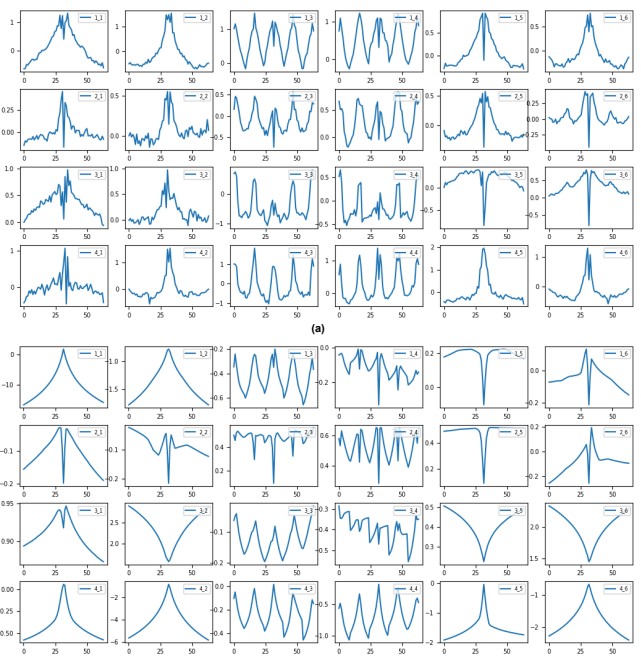

**Figure 2: Comparison between RPB and ES-RPB. The subfigure (a) and (b) showcase the relative position bias (RPB) from the SwinIR model and an GRFomer model where RPB are replaced to ES-RPB, respectively. The subfigure at $i_{th}$ row and $j_{th}$ column corresponds to the relative position bias of $i_{th}$ GRSAB Group and $j_{th}$ GRSAB in the network. These figures specifically highlight the horizontal evolution of the relative position bias values. The x-axis extends from 0 to 62, while the y-axis corresponds to the data taken at the 7th point on the x-axis from an RPB matrix of size 15x63.**

## 2 RELATED WORKS

### 2.1 CNN-Based Image Super-Resolution

A lot of CNN-based SR models [1, 13, 17] have emerged since SR-CNN [6] introduces CNN-based deep learning method for image SR. With very deep convolutional network and residual learning, VDSR [10] achieves a high accuracy for image super-resolution, which deeply influences the subsequent SR models. In order to accelerate SR inference process, FSRCNN [7] learns the mapping from the original low-resolution image to high-resolution and perform an upsampling operation at the end of the network. The pipeline with pixel shuffle upsampling is widely used by subsequent models. While most of other CNN-based SISR methods mainly focus on wider or deeper architecture design, SAN [5] explores more powerful feature expression and feature correlation learning.

### 2.2 Transformer-based Image Super-Resolution

Since Swin Transformer [16] introduces hierarchical architecture and shifted windowing scheme, the feasibility of transformer application in the field of computer vision has been greatly improved.

In order to introduce transformer into the field of image super-resolution, SwinIR[12] is proposed, which has been baseline model for transformer-based SR models. However, for some lightweight scenarios, the amount of parameters and calculations in SwinIR is still too large and its relative position bias lacks a certain organization.

### 2.3 Residuals

Deeper neural networks are more difficult to train, which prevents implementation of deeper networks to improve performance. To ease the difficulty in training of deeper networks, deep residual learning framework is proposed, which helps a large number of subsequent deep models to improve performance. There are a lot of explanations on the reason why residual learning works, among which two are popular: First, residual learning enhance the fitting ability of deep model. Second, residual learning reduces the difficulty of training the model, making it easier to train a better model. Those inspires us to use residual learning framework to ease the difficulty in training of self-attention.

### 2.4 Redundancy of self-attention

Since self-attention [20] is proposed, a large amount of research has been devoted to solving the redundancy of self-attention. After analysis of the empirical similarity of pairwise attention scores across heads and layers, those pairwise attention scores in multiple heads in multiple layers are found to be considerably redundant and Reuse Transformer [2] is proposed to solve the redundancy. Although Reuse Transformer solves the redundancy of self-attention across heads and layers, it fails to pay attention to the redundancy within self-attention. For image super-resolution, SRFormer [22] shrinks the channel dimensions of K and V matrices and then permutes to convey the part of spatial information into the channel dimension, which reduces the redundancy within self-attention while not degenerating too much performance. However, The reduction of the channel dimension of Q and K matrices will make it difficult to add mechanisms such as residuals on this basis. It can be seen from the above discussion that, there is a lot of redundancy within self-attention.

### 2.5 Relative position bias

When self-attention is proposed to solve problems in the field of natural language processing (NLP), absolute position embedding is designed to supplement position information of words in text. After self-attention is applied into image super-resolution, relative position bias enjoys more popularity than the absolute position embedding, because relative position bias can provide the relative position information, which is intrinsically more suitable for image super-resolution. However, there are some problems concerning relative position bias that cannot be ignored, such as parameter redundancy, weak ability to resist interference during training and so on. Afterwards, in order to tackle resolution gaps between pre-training of large vision models, SwinIR-v2 [15] proposes a log-spaced continuous position bias method to effectively transfer large-scale models pre-trained using low-resolution images to downstream tasks with high-resolution inputs. Although the log-spaced continuous position bias is designed to solve the problem of resolution difference

of pre-trained large-scale models, it is very inspiring for the design of relative position bias of transformer-based SR models.

## 3 METHOD

### 3.1 Overall Architecture

The overall architecture of our GRFormer is shown in Fig. 3, consisting of three parts: shallow feature extraction, deep feature extraction, and image reconstruction. Given the LR input $I_{LR} \in R^{H \times W \times C_{in}}$, we first use a 3×3 convolution $L_{SF}$ to transform the low-resolution image $I_{LR}$ to shallow feature $X_0 \in R^{H \times W \times C}$ as

$$X_0 = L_{SF}(I_{LR}) \quad (1)$$

where $C_{in}$ and C is the channel number of LR input and shallow feature. This convolution layer simply converts the input from image space into high-dimensional feature space. Then, we use N grouped residual self-attention block groups $L_{GRSABG}$ and a 3×3 convolution layer $L_{conv}$ at the end to extract the deep feature $I_{DF} \in R^{H \times W \times C}$. The process can be expressed as

$$\begin{aligned} X_i &= L_{GRSABG_i}(X_{i-1}), \\ I_{DF} &= L_{conv}(X_N) + X_0 \end{aligned} \quad (2)$$

In GRSAB Group, given $X_i$ as input, we use M grouped residual self-attention block $L_{GRSAB}$ to get $X_{i,M}$. Then we use a 3×3 convolution layer $L_{conv}$ to get $X_{i+1}$. The process can be expressed as

$$\begin{aligned} X_{i,0} &= X_i, \\ X_{i,j} &= L_{GRSAB_j}(X_{i,j-1}), \\ X_{i+1} &= L_{conv}(X_{i,M}) + X_{i,M} \end{aligned} \quad (3)$$

Finally, we use a 3×3 convolution layer $L_{conv}$ to get better feature aggregation, and aggregate shallow and deep features to reconstruct HR image $I_{HR} \in R^{H \times W \times C_{out}}$ as

$$I_{HR} = L_{shuffle}(L_{conv}(I_{DF}) + I_{SF}), \quad (4)$$

where $C_{out}$ is the channel number of the high-resolution image and $L_{shuffle}$ is a PixelShuffle [18] module.

### 3.2 Grouped Residual Self-Attention Block

The **G**rouped **R**esidual **S**elf-Attention **B**lock (GRSAB) mainly consists of two core components: Grouped Residual Self-Attention (GRSA), Feed Forward Network module (FFN). Given the input of GRSAB as $I_{in} \in R^{H \times W \times C}$, we first use a grouped residual self-attention module $L_{GRSA}$ to learn the global relationships of pixels in a window. After $L_{GRSA}$, we use a LayerNorm module to normalize the feature, because the normalized features can eliminate gradient vanishing and make the training stable. The process can be expressed as

$$I_{GRSA} = Norm(L_{GRSA}(I_{in})) + I_{in} \quad (5)$$

where $I_{GRSA}$ is the ouput of GRSA module. Then we use a feed forward network to transform the $I_{GRSA}$ to another feature space and a LayerNorm module to normalize the feature as

$$I_{out} = Norm(L_{FFN}(I_{GRSA})) + I_{GRSA} \quad (6)$$

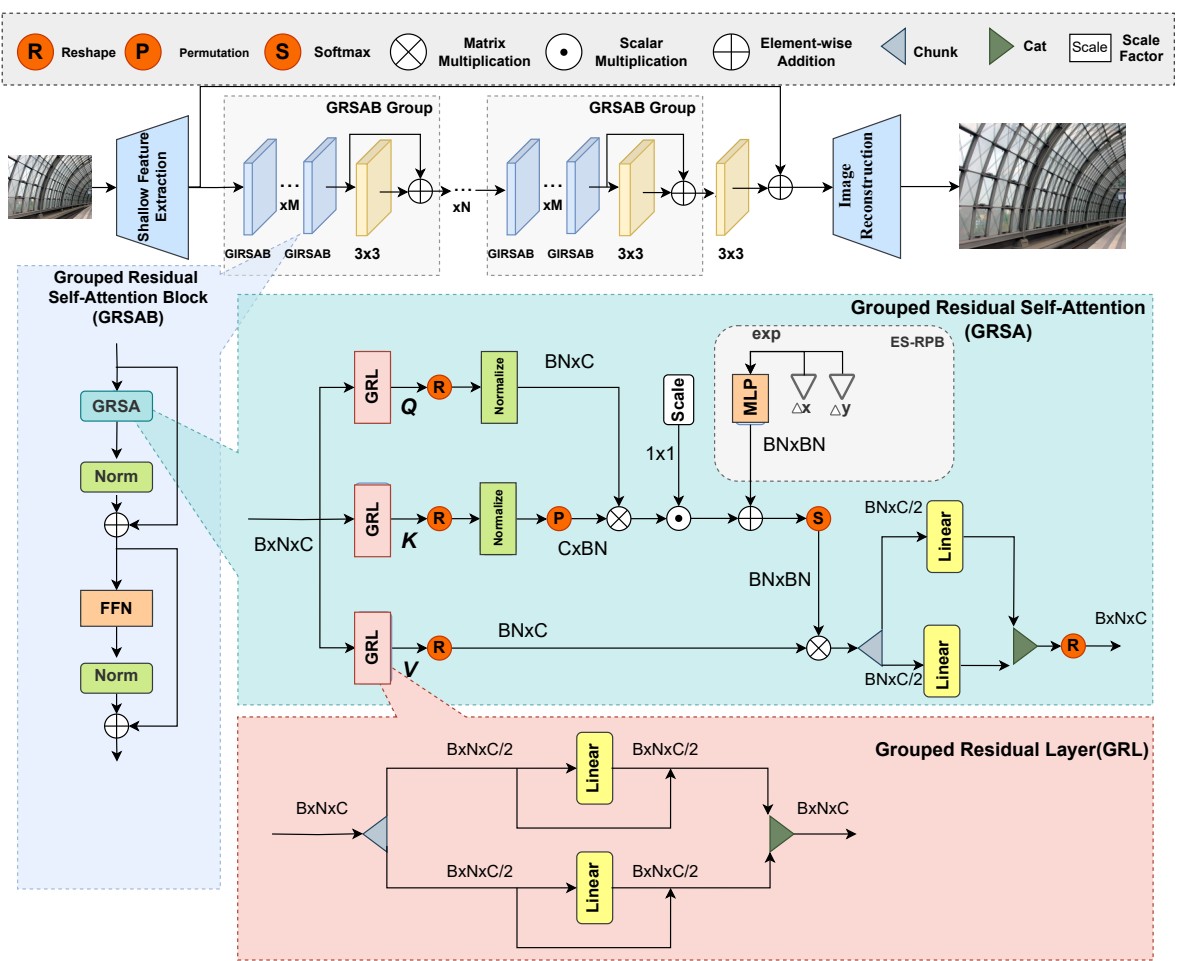

**Figure 3: Network architecture of the proposed GRFormer. It mainly consists of a shallow feature extraction module, several grouped residual self-attention block groups (GRSAB Group) to learn deep feature mapping in an efficient and effective way, and a high-resolution image reconstruction module.**

## 3.3 Grouped Residual Self-Attention

To reduce the number of parameters as well as calculations and enhance the performance, we introduce the Grouped Residual Self-Attention (GRSA), which incorporates two novel and compact components: grouped residual layer (GRL) and exponential-space relative position (ES-RPB). GRL consists of twp parts: residuals for QKV linear layer to transform the feature space of self-attention from the linear space to residual space, grouping scheme for QKV linear layer to reduce the parameters and calculations. By reducing the noise interference during training and making self-attention sensitive to pixel distance, ES-RPB improves the expression ability of position information. The proposed GRSA aggregates features of pixels globally in the window. Specifically, given input of GRSA as $X \in R^{H \times W \times C}$, we uses $L_{GRL_Q}, L_{GRL_K}, L_{GRL_V}$ to get Q, K, V.

$$Q = L_{GRL_Q}(X),$$
$$K = L_{GRL_K}(X), \qquad (7)$$
$$V = L_{GRL_V}(X)$$

where $L_{GRL_Q}, L_{GRL_K}, L_{GRL_V}$ are the GRL module corresponding to Q, K, V. Then we normalize Q and K and calculate the matrix product of Q and K to get the similarity of Q and K. We multiply $QK^T$ by a trainable self-attention scaling factor $\lambda$, add $B_{ES-RPB}$ and perform a Softmax operation. Next, we calculate the matrix product of the $QK^T$ after Softmax and V. If multi-head self-attention is applied, we will use a grouped linear layer $L_{proj}$ as a projection at the last to map the multi-head to one head. The formulation can be written as:

$$\begin{aligned} GRSA = L_{proj}(\text{Softmax}(\lambda \cdot \text{Normalize}(\mathbf{Q}) \\ * \text{Normalize}(\mathbf{K})^\mathbf{T} + \mathbf{B}_{ES-RPB})\mathbf{V}) \end{aligned} \qquad (8)$$

## 3.4 Grouped Residual Linear.

As a substitute for the QKV linear layer, the Grouped Residual Linear (GRL) is one of the core modules of GRSA, with the objective of reducing the amount of parameters and calculations and basically maintaining the feature learning ability. By integrating concepts

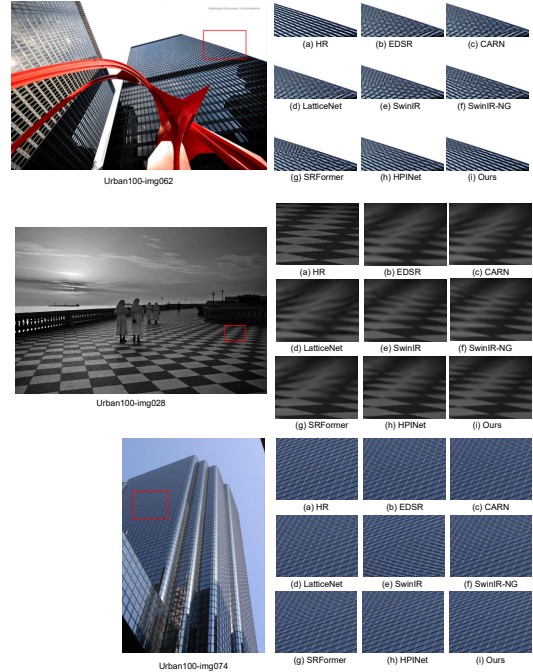

**Figure 4: Qualitative comparison with recent state-of-the-art lightweight image SR methods on the ×4 SR task.**

of *grouping* and *residuals* into the self-attention mechanism, we enhance its structural efficiency and functional effectiveness.

- **Grouping scheme of QKV linear layer in self-attention.** A lot of works, such as [22], show that there is redundancy in generation of Q, K, V and matrix product of Q, K. To reduce the redundancy, we apply the idea of grouping. Given the input as X, we divide X into two equal parts in the channel dimension and then use two independent linear layers to get Q, K, V respectively. Grouping scheme of Q, K, V will not significantly reduce the interaction of pixel features, because the matrix multiplication of Q and K in self-attention will offset these shortcomings to some extent, which is discussed in section 3.6.

- **Residuals of the Q, K, V linear layers.** Residuals allows training to be performed in the residual space, which makes the network find the optimal solution in the residual space. So we add it into the QKV linear layer to transform the training space of QKV linear layer from linear space to residual space in order to enhance the feature learning ability of QKV linear layer.

Specifically, we assume the input of GRL module is $X \in R^{H \times W \times C}$. We first divide X in channel dimension into two parts $X_{in_1}, X_{in_2}$. Then, by using residuals, each uses a linear layer to get $X_{out_1}, X_{out_2}$. Finally, we merge $X_{out_1}, X_{out_2}$ in channel dimension to get the output $X_{out}$ at the last as

$$
\begin{aligned}
X_{in_1}, X_{in_2} &= X, \\
X_{out_1} &= L_{Linear_1}(X_{in_1}) + X_{in_1}, \\
X_{out_2} &= L_{Linear_2}(X_{in_2}) + X_{in_2}, \\
X_{out} &= X_{out_1}, X_{out_2},
\end{aligned}
\tag{9}
$$

where $L_{Linear_1}$ and $L_{Linear_2}$ are linear layers, $X_{in_1}, X_{in_2}, X_{out_1}, X_{out_2} \in R^{H \times W \times \frac{C}{2}}$.

## 3.5 Exponential-Space Relative Position Bias

To solve the four disadvantages of original RPB mentioned in **RQ3**, we propose the Exponential-Space Relative Position Bias (ES-RPB). We design a exponential mapping for original absolute position coordinates to forcibly add pixel distance sensitive rules to original RPB, which makes it give more weight to nearby pixels. What's more, we use a tiny multilayer perceptron (MLP) to obtain the mapping of all absolute position coordinates, which reduces the impact of noise during training and makes it easier to be trained. Specifically, we transform the abscissa ΔX and the ordinate ΔY in absolute position coordinate matrix from linear space to exponential space, and then we use a tiny MLP to get $B_{ES-RPB}$

$$
\begin{aligned}
\Delta \widehat{X} &= sign(\Delta X) * (1 - \exp(-|\alpha * \Delta X|)), \\
\Delta \widehat{Y} &= sign(\Delta Y) * (1 - \exp(-|\beta * \Delta Y|)), \\
B_{ES-RPB} &= MLP(\Delta \widehat{X}, \Delta \widehat{Y})
\end{aligned}
\tag{10}
$$

where $\alpha$ and $\beta$ is trainable distance-sensitive factors to control the sensitivity to distances between reference pixel and the others in the same window and the symbol * represents the multiplication of each position in the matrix. MLP consists of two linear layers and an activation layer sandwiched between them.

## 3.6 Explanation of the effectiveness of GRL

GRL consists of two parts: a grouping scheme and a residual structure. Grouping scheme is proposed to efficiently reduce the parameters and calculations without severe performance degradation, and residual structure makes the network find the optimal solution in the residual space, thereby reducing the difficulty of feature learning. The explanation is as follows:

- **Explanation of the effectiveness of grouping scheme.** First, we analyse the effectiveness in reduction of parameters and MACs. We assume that the number of input features is N. Then, both the parameters and the MACs occupied by the linear layer are $N^2$. But, if we group the input features into halves and use two linear layers with the number of input features of $\frac{N}{2}$ to process the two parts of input features, respectively, both parameters and MACs occupied by the two linear layers are only $\frac{N^2}{2}$. Obviously, the grouping scheme can halve the number of parameters and MACs.

Second, we analyse the effectiveness analysis in preventing severe performance degradation. A possible concern about the grouping scheme is that grouping features will result in a lack of aggregation of the two groups of features, leading to performance degradation. We will analyze below that, at least for Q, K matrices, this worry is unnecessary. As shown in the self-attention formula: $\text{Attention}(Q, K, V) = \text{Softmax}\left(\frac{QK^T}{\sqrt{d_k}}\right) V$, Q and K matrices are only used to perform matrix product and get $QK^T$ matrix. During the matrix product of Q and K, even if we employ the grouping scheme to divide the input features into two groups and perform feature aggregation individually, the subsequent

**Table 1: Quantitative evaluations of the lightweight GRFormer against state-of-the-art methods on commonly used benchmark datasets. Best and second best results are marked in red and blue colors. #Params means the number of the network parameters. #MACs denotes the number of the MACs which are calculated on images with an upscaled spatial resolution of 1280 × 720 pixels. #Weighted-Avg means the weighted average PSNR and SSIM on five benchmark datasets. #Weighted-Avg = $\sum_{i=1}^{5} M_i \times Score_i$, where M is the number of images in the dataset and Score is the corresponding PSNR or SSIM score.**

| Scale | Method | Year | #Params(/K) | #MACs(/G) | Set5 | Set14 | B100 | Urban100 | Manga109 | #Weighted-Avg |
|---|---|---|---|---|---|---|---|---|---|---|
| ×2 | EDSR-baseline[13] | CVPRW2017 | 1370 | 316.3 | 37.99/0.9604 | 33.57/0.9175 | 32.16/0.8994 | 31.98/0.9272 | 38.54/0.9769 | 34.37/0.9353 |
| | CARN [1] | ECCV2018 | 1592 | 222.8 | 37.76/0.9590 | 33.52/0.9166 | 32.09/0.8978 | 31.92/0.9256 | 38.36/0.9765 | 34.27/0.9342 |
| | LatticeNet [17] | ECCV2020 | 756 | 169.7 | 38.15/0.9610 | 33.78/0.9193 | 32.25/0.9005 | 32.43/0.9302 | -/- | -/- |
| | SwinIR-light [12] | ICCV2021 | 910 | 207.5 | 38.14/0.9611 | 33.86/0.9206 | 32.31/0.9012 | 32.76/0.9340 | 39.12/0.9783 | 34.87/0.9386 |
| | SwinIR-NG [4] | CVPR2023 | 1181 | 210.7 | 38.17/0.9612 | 33.94/0.9205 | 32.31/0.9013 | 32.78/0.9340 | 39.20/0.9781 | 34.90/0.9385 |
| | DLGSANet-light [11] | ICCV2023 | 745 | 169.4 | 38.20/0.9612 | 33.89/0.9203 | 32.30/0.9012 | 32.94/0.9355 | 39.29/0.9780 | 34.98/0.9389 |
| | SRFormer-light [22] | ICCV2023 | 853 | 198.6 | 38.23/0.9613 | 33.94/0.9209 | 32.36/0.9019 | 32.91/0.9353 | 39.28/0.9785 | 34.98/0.9393 |
| | HPINet-M [14] | AAAI2023 | 783 | 213.1 | 38.12/- | 33.94/- | 32.31/- | 32.85/- | 39.08/- | 34.88/- |
| | **GRFormer (Ours)** | - | 781 | 198.4 | 38.22/0.9614 | 34.01/0.9214 | 32.35/0.9018 | 33.17/0.9375 | 39.30/0.9785 | 35.07/0.9399 |
| ×3 | EDSR-baseline[13] | CVPRW2017 | 1555 | 160.1 | 34.37/0.9270 | 30.28/0.8417 | 29.09/0.8052 | 28.15/0.8527 | 33.45/0.9439 | 30.38/0.8692 |
| | CARN [1] | ECCV2018 | 1592 | 118.6 | 34.29/0.9255 | 30.29/0.8407 | 29.06/0.8034 | 28.06/0.8493 | 33.50/0.9440 | 30.36/0.8676 |
| | LatticeNet [17] | ECCV2020 | 765 | 76.2 | 34.53/0.9281 | 30.39/0.8424 | 29.15/0.8059 | 28.33/0.8538 | -/- | -/- |
| | SwinIR-light [12] | ICCV2021 | 918 | 94.2 | 34.62/0.9289 | 30.54/0.8463 | 29.20/0.8082 | 28.66/0.8624 | 33.98/0.9478 | 30.76/0.8746 |
| | SwinIR-NG [4] | CVPR2023 | 1190 | 95.67 | 34.64/0.9293 | 30.58/0.8471 | 29.24/0.8090 | 28.75/0.8639 | 34.22/0.9488 | 30.89/0.8757 |
| | DLGSANet-light [11] | ICCV2023 | 752 | 75.8 | 34.70/0.9295 | 30.58/0.8465 | 29.24/0.8089 | 28.83/0.8653 | 34.16/0.9483 | 30.89/0.8759 |
| | SRFormer-light [22] | ICCV2023 | 861 | 88.3 | 34.67/0.9296 | 30.57/0.8469 | 29.26/0.8099 | 28.81/0.8655 | 34.19/0.9489 | 30.90/0.8764 |
| | HPINet-M [14] | AAAI2023 | 924 | 110.6 | 34.70/- | 30.63/- | 29.26/- | 28.93/- | 34.21/- | 30.95/- |
| | **GRFormer (Ours)** | - | 789 | 93.5 | 34.67/0.9293 | 30.64/0.8481 | 29.27/0.8100 | 29.07/0.8702 | 34.35/0.9494 | 31.04/0.8781 |
| ×4 | EDSR-baseline[13] | CVPRW2017 | 1518 | 114.2 | 32.09/0.8938 | 28.58/0.7813 | 27.57/0.7357 | 26.04/0.7849 | 30.35/0.9067 | 28.14/0.8119 |
| | CARN [1] | ECCV2018 | 1592 | 90.88 | 32.13/0.8937 | 28.60/0.7806 | 27.58/0.7349 | 26.07/0.7837 | 30.47/0.9084 | 28.19/0.8118 |
| | LatticeNet [17] | ECCV2020 | 777 | 43.6 | 32.30/0.8962 | 28.68/0.7830 | 27.62/0.7367 | 26.25/0.7873 | -/- | -/- |
| | SwinIR-light [12] | ICCV2021 | 930 | 54.18 | 32.44/0.8976 | 28.77/0.7858 | 27.69/0.7406 | 26.47/0.7980 | 30.92/0.9151 | 28.51/0.8204 |
| | SwinIR-NG [4] | CVPR2023 | 1201 | 55 | 32.44/0.8980 | 28.83/0.7870 | 27.73/0.7418 | 26.61/0.8010 | 31.09/0.9161 | 28.62/0.8221 |
| | DLGSANet-light [11] | ICCV2023 | 761 | 43.2 | 32.54/0.8993 | 28.84/0.7871 | 27.73/0.7415 | 26.66/0.8033 | 31.13/0.9161 | 28.65/0.8227 |
| | SRFormer-light [22] | ICCV2023 | 873 | 53 | 32.51/0.8988 | 28.82/0.7872 | 27.73/0.7422 | 26.67/0.8032 | 31.17/0.9165 | 28.67/0.8230 |
| | HPINet-M [14] | AAAI2023 | 896 | 81.1 | 32.60/- | 28.87/- | 27.73/- | 26.71/- | 31.19/- | 28.69/- |
| | **GRFormer (Ours)** | - | 800 | 50.8 | 32.58/0.8994 | 28.88/0.7886 | 27.75/0.7431 | 26.90/0.8097 | 31.31/0.9183 | 28.80/0.8260 |

matrix product of Q and K will deeply aggregate the two groups of features. The process can be shown as follows.

Specifically, given input of self-attention as X ∈ $R^{N \times C}$, where N is the number of pixels in a window and C is number of input features, we cut up X into 2×2 blocks as

$$\begin{pmatrix} X_{11} & X_{12} \\ X_{21} & X_{22} \end{pmatrix} = X, \tag{11}$$

where $X_{11}, X_{12}, X_{21}, X_{22} \in R^{\frac{N}{2} \times \frac{C}{2}}$. By packing parameter matrices of two grouped linear layers with input features of $\frac{C}{2}$ into one parameter matrix, we can assume the parameter matrices of Q, K projections as $\begin{pmatrix} M_{Q_1} \\ M_{Q_2} \end{pmatrix}$, $\begin{pmatrix} M_{K_1} \\ M_{K_2} \end{pmatrix}$ respectively, where $M_{Q_1}, M_{Q_2}, M_{K_1}, M_{K_2} \in R^{\frac{C}{2} \times \frac{C}{2}}$. Then we can perform matrix product of X and the parameter matrices of Q, K projections respectively as

$$\begin{aligned} Q, K &= \begin{pmatrix} X_{11} & X_{12} \\ X_{21} & X_{22} \end{pmatrix} * \begin{pmatrix} M_{Q_1 \vee K_1} \\ M_{Q_2 \vee K_2} \end{pmatrix} \\ &= \begin{pmatrix} X_{11} * M_{Q_1 \vee K_1} & X_{12} * M_{Q_2 \vee K_2} \\ X_{21} * M_{Q_1 \vee K_1} & X_{22} * M_{Q_2 \vee K_2} \end{pmatrix}, \end{aligned} \tag{12}$$

where ∨ represents the logical symbol "or". Then, we perform the matrix product of Q and K as follows:

$$QK^T = \begin{pmatrix} O_{11} & O_{12} \\ O_{21} & O_{22} \end{pmatrix}, \tag{13}$$

then, each element in matrix of $QK^T$ is calculated:

$$\begin{aligned} O_{11} &= X_{11} M_{Q_1} M_{K_1}{}^T X_{11}{}^T + X_{12} M_{Q_2} M_{K_2}{}^T X_{12}{}^T, \\ O_{12} &= X_{11} M_{Q_1} M_{K_1}{}^T X_{21}{}^T + X_{12} M_{Q_2} M_{K_2}{}^T X_{22}{}^T, \\ O_{21} &= X_{21} M_{Q_1} M_{K_1}{}^T X_{11}{}^T + X_{22} M_{Q_2} M_{K_2}{}^T X_{12}{}^T, \\ O_{22} &= X_{21} M_{Q_1} M_{K_1}{}^T X_{21}{}^T + X_{22} M_{Q_2} M_{K_2}{}^T X_{22}{}^T \end{aligned} \tag{14}$$

It can be seen from Equation 14, in the matrix product of Q and K of the self-attention mechanism, both $X_{11}$ and $X_{12}$ are used to perform a series of matrix products to get $O_{11}$, $O_{12}$, and so do $X_{21}$ and $X_{22}$ to get $O_{21}$, $O_{22}$, which aggregates two groups of features. Therefore, the matrix product of Q and K will offset the weak aggregation of input features brought by the use of grouping scheme. In other words, our grouping scheme won't lead to lack of aggregation of the two groups of features and thereby won't lead to severe performance degradation.

- **Explanation of the effectiveness of residual structure for the QKV linear layer.** As shown in Formula 8, self-attention involves multiple matrix products, which makes it difficult for the network to learn the optimal parameters. Given the input of self-attention as X and the optimal Q, K, V as $\widehat{Q}, \widehat{K}, \widehat{V}$, if we add the residual structure for QKV linear layer, the network only needs to optimize $\widehat{Q}$-X, $\widehat{K}$ - X, $\widehat{V}$ - X instead of $\widehat{Q}, \widehat{K}, \widehat{V}$. The residual structure transforms the learning space of QKV linear layer from linear space to residual space, enhancing the feature learning ability of QKV linear layer.

## 3.7 Explanation of the effectiveness of ES-RPB

We mainly analyze the effectiveness of ES-RPB from the perspective of parameter reduction and performance improvement for ES-RPB.

- **Explanation of parameter reduction for ES-RPB.** We assume that the size of the window of self-attention is W×H and the self-attention head is 1. The number of parameters occupied by the original RPB is (2×W-1)×(2×H-1). Specifically, the window size is usually set to 16×16, so the parameter count is 961 in this case. In contrast, if we suppose that the number of features of the hidden layer in MLP is $C_{hidden}$, the amount of parameters occupied by ES-RPB is $3C_{hidden}$. Specifically, $C_{hidden}$ is set to 128 in our GRFormer, so the number of parameters occupied by ES-RPB in GRFormer is 384, which is much less than 961. What has to be noted is that, when the height and width of window grow simultaneously at a linear rate, the parameters occupied by original RPB will grow squarely, while the parameters occupied by ES-RPB will not grow.
- **Explanation of performance improvement for ES-RPB.** The ES-RPB mechanism within GRFormer improves performance through three key strategies. Firstly, instead of training positional parameters directly, which can be noise-sensitive, we utilize a tiny MLP to get the $B_{ES-RPB}$. This approach minimizes the noise impact on positional parameters during training. Secondly, the mechanism enhances interaction of parameters representing relative position information by training them through this tiny MLP rather than in isolation. Thirdly, ES-RPB introduces a distance-sensitive design. It employs an exponential function to map the absolute positional coordinates $(\Delta X, \Delta Y)$ from linear space to exponential space, resulting in $\Delta \widehat{X}$ and $\Delta \widehat{Y}$. This transformation ensures that positions closer to the reference pixel exhibit more significant changes, aligning with the principle that nearer pixels should attract more attention.

## 4 EXPERIMENTS

In this section, we conduct experiments on the lightweight image SR tasks, compare our GRFormer with existing state-of-the-art methods, and do ablation analysis of the proposed method.

## 4.1 Experimental Setup

**Datasets and Evaluation**. For training, we use DIV2K (Agustsson and Timofte 2017), the same as the comparison models, to train our GRFormer. It includes 800 training images and 100 validation images, mainly concerning human, animals, plants, buildings, etc. For testing, we use five public SR benchmark datasets: Set5 (Bevilacqua et al. 2012), Set14 (Zeyde, Elad, and Protter 2010), B100 (Martin et al. 2001), Urban100 (Huang, Singh, and Ahuja 2015) and Manga109 (Matsui et al. 2017) to evaluate model. The experimental results are evaluated in terms of two objective criteria: peak signal-to-noise ratio (PSNR) and structural similarity index (SSIM), which are both calculated on the Y channel from the YCbCr space.

**Implementation Details.** We set the GRSAB Group number, GRSAB number of a GRSAB Group, feature number, and attention head number, window size to 4, 6, 60, 3, 8×32, respectively. The training low-resolution patch size we use is 64×64. When training, we randomly rotate the images by 0°, 90°, 180°, 270° and randomly flip images horizontally for data augmentation. We adopt the Adam

optimizer with $\beta_1 = 0.9$ and $\beta_2 = 0.99$ to train the model for 600k iterations. The learning rate is initialized as $2 \times 10^{-4}$ and halves on {250000, 400000, 510000, 540000}-th iterations. We use L1 loss to train the model. The whole process is implemented by Pytorch on NVIDIA GeForce RTX 3080 GPUs.

## 4.2 Comparisons with State-of-the-arts

We compare our GRFormer with commonly used lightweight SR models for upscaling factor ×2, ×3, ×4, including EDSR [13], CARN[1], LatticeNet[17], SwinIR[12], SwinIR-NG[4], HPINet[14], DLGSANet[11], SRFormer[22]. We compare the parameters, calculations as well as performance on five commonly used SR benchmark datasets (Set5, Set14, B100, Urban100, Manga109). The comparison results are grouped for ×2, ×3, ×4 upscaling factor.

**Quantitative Comparison** Table 1 shows quantitative comparisons in terms of PSNR and SSIM on five benchmark datasets. As shown in Table 1, our GRFormer achieves the best PSNR score and SSIM score for ×2, ×3, ×4 task on Set14, Urban100, Manga109 and weighted average of the five benchmark datasets, and the PSNR score and SSIM score achieved by our GRFormer is either quite close or superior to that of SOTA model on Set5 and B100. What's more, GRFormer outperforms SOTA model on Urban100 and Manga109 by a large margin. It is worth noting that, our GRFormer outperforms SwinIR-light by a maximum PSNR of 0.42dB and SOTA by a maximum PSNR of 0.23dB, which is a significant improvement for image SR. Furthermore, our GRFormer outperforms SOTA model by about 0.1dB on the weighted average of the five benchmark datasets for ×2, ×3, ×4 task. Although our GRFormer achieves great performance, the parameters and MACs of GRFormer is relatively low. As shown in Fig. 1 (a), compared with the self-attention of other transformer-based SR models, our GRSA has the smallest number of parameters and calculations.

**Qualitative Comparison** We further show visual examples of common used methods under scaling factor ×4. As shown in Fig.4, we use three images reconstructed by EDSR, CARN, LatticeNet, SwinIR, SwinIR-NG, SRFormer, HPINet and GRFormer to make qualitative comparisons.

First, we make qualitative comparisons on Urban100-img062. We can see that, the texture and color of the image reconstructed by EDSR[13], CARN[1] and LatticeNet[17] are distorted, and the lower right corner of the restored image is severely distorted. SwinIR, SwinIR-NG and SRFormer reconstruct part of the texture well, but there are still large areas with severe distortion. The relatively large area of image texture reconstructed by HPINet is distorted. The image reconstructed by our GRFormer has the smallest distortion area, and the restored color is also closest to HR.

Second, we make qualitative comparisons on Urban100-img028. Urban100-img028 is an image of the ground whose texture regularly changes from large to small. The distorted and blurred areas of the image reconstructed by EDSR, CARN, LatticeNet and SRFormer are visibly large. For the middle area of the image, the image reconstructed by SwinIR is relatively blurry and the image reconstructed by HPINet is slightly blurry. There is relatively large deformation in the image reconstructed by SwinIR-NG. Apparently, compared with other models, the picture quality recovered by GRFormer is the

**Table 2: Effect of the GRFormer on SISR. SA-Params and SA-MACs mean the parameters and MACs in our GRSA respectively. Params and MACs mean the parameters and MACs in our GRFormer. The ablation experiments are trained on DF2K for ×4 SR task and tested on benchmark datasets (Set5, Set14, B100, Urban100, Manga109) to get PSNR and SSIM.**

| Model | GRL | | ES-RPB | #SA-Params | #Params | #SA-MACs | #MACs | Set5 | Set14 | B100 | Urban100 | Manga109 |
|---|---|---|---|---|---|---|---|---|---|---|---|---|
| | Group | Residuals | | | | | | | | | | |
| ① | ✓ | ✓ | ✓ | 8.2K | 810K | 39.5M | 50.8G | 32.59/0.8999 | 28.92/0.7890 | 27.77/0.7435 | 26.97/0.8110 | 31.47/0.9195 |
| ② | ✓ | ✓ | | 10.3K | 850K | 38.9M | 50.8G | 32.46/0.8980 | 28.81/0.7865 | 27.72/0.7414 | 26.53/0.7998 | 31.07/0.9150 |
| ③ | | | ✓ | 15.4K | 973K | 69M | 61.4G | 32.60/0.9000 | 28.93/0.7892 | 27.77/0.7437 | 27.00/0.8118 | 31.48/0.9197 |
| ④ | | ✓ | ✓ | 15.4K | 973K | 69M | 61.4G | 32.63/0.9002 | 28.95/0.7897 | 27.78/0.7441 | 27.07/0.8139 | 31.56/0.9206 |
| ⑤ | ✓ | | ✓ | 8.2K | 810K | 39.5M | 50.8G | 32.58/0.8996 | 28.91/0.7888 | 27.76/0.7434 | 26.97/0.8116 | 31.43/0.9191 |

best and GRFormer has good reconstruction effect on a relatively small texture scale.

Third, we make qualitative comparison on Urban100-img074. The direction of the window frame in Urban100-img074 (a) is clear, which can help us easily distinguish whether it is distorted. It can be easily seen that except GRFormer, the image reconstructed by all other methods are more or less distorted. Obviously, our GRFormer reconstructs the image very well in various details, which shows the superiority of our method.

## 4.3 Ablation Analysis

We conduct ablation experiments to study the effect of Grouped Residual Layer (GRL) and Exponential-Space Relative Position Bias (ES-RPB). Ablation experiments are trained on DF2K and evaluated on the Set5, Set14, B100, Urban100, Manga109 datasets. PSNR and SSIM are adopted to evaluate the perceptual quality of recovered images. We also adopt parameters and MACs on images with an upscaled spatial resolution of $1280 \times 720$ pixels to evaluate the complexity.

Specifically, ablation experiments are conducted as follows. First, we start with a complete model with GRL and ES-RPB (model ①). Second, we replace the GRL of model ① with a linear layer to get model ③. Third, we replace the ES-RPB of model ① with RPB in SwinIR [12] to obtain model ②. Finally, to prove the effectiveness of the grouping scheme and residual structure in GRL, we retain the ES-RPB and remove the grouping scheme and residual structure separately to obtain model ④ and model ⑤ respectively. The results are shown in Table 2.

**Effectiveness of GRL** As shown in Table 2, compared with model ③, #SA-Params, #Params, #SA-MACs and #MACs of model ① reduce by 47%, 17%, 43%, 17% respectively. A significant reduction in #SA-Params and #SA-MACs can be seen, because our methods mainly act on self-attention. Although the number of parameters and MACs are significantly reduced, application of GRL barely degrades performance, which shows the superiority of GRL. To further show the effectiveness of grouping scheme and residual structure in GRL, we conduct further experiments to get the model ④ and model ⑤ respectively.

- Effectiveness of grouping scheme. Compared with model ④, #SA-Params, #Params, #SA-MACs and #MACs of model ① reduce by 47%, 17%, 43%, 17% respectively, but model ① suffers some performance degradation, especially on Urban100 as well as Manga109. However, our residual structure for QKV linear layer can greatly reduce performance degradation.
- Effectiveness of residual structure. Compared with model ③, model ④ doesn't make any changes to #SA-Params, #Params,

#SA-MACs and #MACs, but significantly improves the performance, especially on Urban100 and Manga109. Specifically, the performance of model ④ improves by 0.07dB on Urban100 and 0.08dB on Manga109 respectively.

**Effectiveness of ES-RPB** The core feature of ES-RPB is the ability to represent the pixel position information. To highlight the contribution of our ES-RPB, we replace the ES-RPB of model ① with RPB in SwinIR to get model ②. As shown in Table 2, compared with model ②, model ① improves the performance on five benchmark datasets in a large margin. Specifically, model ① outperforms model ② on Urban100 by 0.44dB PSNR score, which is a notable boost in lightweight image super-resolution. To further understand the reason of improvement brought by ES-RPB, we draw the three dimensional view of both ES-RPB in model ① and RPB in model ②. As shown in Fig.2, we can see that, the curve of ES-RPB in model ① is roughly similar to that of RPB in model ②, because both ES-RPB and RPB represent the relative position information. As shown in Fig.2, after comparing the curves of RPB and ES-RPB, we can find two differences: First, minor fluctuations on the curve of ES-RPB are much less than that of RPB, which means that most of noise interference is removed. Secondly, we can see from the curve of RPB that RPB is overfitted, which will lead to poor generalization ability. In contrast, our ES-RPB use simpler structure to improve generalization ability. It means that RPB learns a lot of content that is not universally applicable, which affects its generalization ability to represent relative position information.

## 5 CONCLUSION

In this paper, we propose GRSA, an efficient self-attention mechanism which consists of two components: GRL to significantly reduce the amount of parameters and calculations as well as ES-RPB to efficiently and effectively represent the relative position information and make it distance-sensitive. Within GRL module, we use grouping scheme to reduce redundancy in terms of parameters as well as calculations with as little performance degradation as possible and residual structure to enhance feature learning ability for QKV linear layer. Based on GRSA, we design a simple yet effective model for lightweight single image super-resolution, called GRFormer. Benefiting from GRL and ES-RPB, GRFormer not only significantly reduces the number of parameters and MACs, but also enhances the performance in terms of PSNR and SSIM. Experimental results show the superior performance of GRFormer over previous state-of-the-art lightweight SR models on benchmark datasets, especially on Urban100 and Manga109. We hope our GRSA can serve as a useful tool for future research on the design of SR models.

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
