# OpenReview forum: "GRFormer: Grouped Residual Self-Attention for Lightweight Single Image Super-Resolution"
_acmmm.org/ACMMM/2024/Conference — MM2024 Poster_

### Official Review · Reviewer_Ag2u · 2024-04-29

**Rating:** 4
**Confidence:** 4

**Summary:**

The paper introduces GRFormer, a lightweight image super-resolution (SR) model. The authors designed the Grouped Residual Self-Attention (GRSA), an efficient self-attention mechanism. GRSA leverages a Grouped Residual Layer (GRL) to reduce the computational complexity of linear operations and an Exponential Space Relative Position Bias (ES-RPB) as a replacement for RPB. Experiments demonstrate that the proposed method achieves better SR performance.

**Strengths:**

1. The introduction of Grouped Residual Self-Attention (GRSA), utilizing a Grouped Residual Layer (GRL), reduces complexity and parameter count while maintaining performance.
2. The design of the ES-RPB is well-suited for IR tasks.
3. The method show better performance when compared with existing lightweight approaches.

**Limitations:**

1. The paper does not provide concrete comparisons of running times. Although the proposed GRL and ES-RPB reduce the number of parameters and computational complexity, operations such as split operation, residual connections, and MLP significantly increase runtime, and the first two operations do not add to the computational complexity. Thus, merely reducing parameters and FLOPs is of limited significance.
2. In the ablation studies, there is a lack of a baseline without both GRL and ES-RPB. Additionally, the baseline used in the paper is not clearly defined. The method section indicates the use of global SA in GRSA, which contradicts the low computational complexity. Therefore, clarification is needed on the specific self-attention variant used in GRFormer.
3. Typographical error: '285 Swin Transformer v2' (L285, SwinIR-v2).
4. Increased citations are needed; the paper compares various Transformer-based methods in experimental sections but should also expand the related work section to include these comparisons.

**Suitability:**

3

---

### Official Review · Reviewer_kmiL · 2024-05-05

**Rating:** 3
**Confidence:** 3

**Summary:**

This paper proposed Grouped Residual Self-Attention Transformer (GRFormer) for lightweight image SR. The authors primarily focus on enhancing the self-attention mechanism by introducing two key components: the Grouped Residual Layer (GRL) and Exponential-Space Relative Position (ES-RPB).  The proposed methods show comparable performance with previous advanced lightweight SR methods.

**Strengths:**

1. The proposed ES-RPB effectively reduced the noise and parameters between positional embedding bias, and the authors provided good visualized examples to further illustrate its effectiveness. This insight is interesting and novel.
2. The overall writing is clear and easy to follow, facilitating comprehension for readers.
3. The visual comparison is good.

**Limitations:**

1. The overall improvements are relatively minor compared with the second best, especially in SR $\times2$ and $\times3$.
2. The proposed methods are only conducted on lightweight SR methods, however more solid methods e.g., ELAN (ECCV 2022) demonstrate its effectiveness on both lightweight and large-scale tasks. Additionally, the ablation study of the proposed methods are not adequate, so the overall workflow is not solid for this paper.
3. The window size is $8\times32$, and this will contribute a lot to the overall performance but not equal with previous methods.


Minor points:
1. The motivation behind the proposed GRL remains ambiguous. It is unclear why the feature is partitioned into two segments, undergoes linear transformation, and is subsequently concatenated together.
2. $\times$ and $x$ are not consistent in the manuscript.
3. QKV linear layer is better to declare as Query, Key, and Value
(QKV) linear layers in the beginning.

**Suitability:**

2

---

### Official Review · Reviewer_qA42 · 2024-05-18

**Rating:** 2
**Confidence:** 4

**Summary:**

This paper introduces GRFormer, an efficient and lightweight single-image super-resolution (SISR) method. GRFormer significantly reduces parameter overhead and computation while enhancing performance by incorporating a Grouped Residual Self-Attention mechanism (GRSA). Extensive experiments are conducted to show the effectiveness of the proposed approach.

**Strengths:**

1. This paper proposes an effective  lightweight model GRFormer for image super-resolution.
2. The proposed method achieves notable performance improvements compared to previous work.

**Limitations:**

1. The writing quality of this paper is insufficient, with an excessive amount of textual description and a lack of relevant visual experiments to demonstrate the effectiveness of the proposed method. For instance, it is unclear whether the proposed modifications actually reduce redundancy in self-attention. Additionally, there are many issues with the citation of references.
2. The content and innovativeness of this paper are insufficient, as it essentially makes only minor modifications to the extraction of QKV and positional encoding, and it focuses solely on the task of lightweight super-resolution.
3. The ablation experiments are incomplete and somewhat disorganized. Table 2 lacks a baseline, and it is unclear why the ablation experiments are conducted using the DF2K dataset.
4. The model uses a larger window size (i.e., 8×32), and the performance improvement might be primarily due to the increase in window size rather than the proposed methods [1]. This has not been clarified through ablation studies.
5. For the task of lightweight super-resolution, comparing the increase in model inference speed, rather than just the number of parameters and MACs, might better reflect the lightweight nature of the model.

[1] SRFormer: Permuted Self-Attention for Single Image Super-Resolution, ICCV2023.

**Suitability:**

2

---

### Official Review · Reviewer_pN92 · 2024-05-22

**Rating:** 3
**Confidence:** 4

**Summary:**

This paper designs a lightweight single-image super-resolution model, called GRFormer, and proposes a novel self-attention mechanism, GRSA, which consists of two core components: the GRL and the ES-RPB. The GRL reduces parameter and computational redundancy through a grouping scheme, and enhances the feature learning capability of the linear layer of the QKV through the residual structure. ES-RPB is used to represent relative position information effectively and efficiently.

**Strengths:**

1. The proposed ES-RPB achieves higher performance with a smaller number of parameters in a simple and effective manner.
2. This paper is well-organized, clearly explains the motivations, and analyzes the reasonableness of the methodology.

**Limitations:**

1. The experiment is inadequate. In the introduction, it is claimed that “the GRSA module can seamlessly replace the self-attention module and its variants in other transformer-based SR models”. Relevant experiments are necessary to demonstrate the potential of the GRSA module.

2. Grouped convolution has been proven to reduce the number of operations and parameters and has been widely used in various lightweight models. The authors should further compare it with existing methods and clarify the unique innovations of the proposed method.

3.Some small mistakes:
a) “To sum up,our contributions can be summarized as follows:” should be “To sum up, our”
b) “GRL consists of twp parts” should be “GRL consists of two parts”
c) Figure 1 should indicate what each axis represents.

**Suitability:**

2

---

### Meta-Review · Area_Chair_mEru · 2024-07-02

**Recommendation:** Accept (Poster)
**Confidence:** 5

**Metareview:**

Given the significant performance improvements demonstrated by GRFormer but also considering the areas requiring further clarification and expansion, the paper currently stands at a borderline accept. Addressing the aforementioned recommendations will significantly enhance the paper's quality and may lead to a more favorable evaluation in future submissions.